# BANANA: A BENCHMARK FOR THE ASSESSMENT OF NEURAL ARCHITECTURES FOR NUCLEIC ACIDS

## ABSTRACT

Machine learning has always played an important role in bioinformatics and recent applications of deep learning have allowed solving a new spectrum of biologically relevant tasks. However, there is still a gap between the "mainstream" AI and the bioinformatics communities. This is partially due to the format of bioinformatics data, which are typically difficult to process and adapt to machine learning tasks without deep domain knowledge. Moreover, the lack of standardized evaluation methods makes it difficult to rigorously compare different models and assess their true performance. To help to bridge this gap, and inspired by work such as SuperGLUE and TAPE, we present BANANA, a benchmark consisting of six supervised classification tasks designed to assess language model performance in the DNA and RNA domains. The tasks are defined over three genomics and one transcriptomics languages (human DNA, bacterial 16S gene, nematoda ITS2 gene, human mRNA) and measure a model's ability to perform whole-sequence classification in a variety of setups. Each task was built from readily available data and is presented in a ready-to-use format, with defined labels, splits, and evaluation metrics. We use BANANA to test state-of-the-art NLP architectures, such as Transformer-based models, observing that, in general, self-supervised pretraining without external corpora is beneficial in every task.

## 1 INTRODUCTION

Since the advent of sequencing techniques (Sanger et al., 1977), there has been an exponentially increasing availability of data in the form of sequences of DNA, RNA, and proteins. Analysis of these sequences has marked many milestones in every field of biology, from the Human Genome Project (Watson, 1990) to the engineering of vaccines for the SARS-COV-2 virus (Chen et al., 2020), from climate-change studies (Jansson & Hofmockel, 2020) to forensics (Bianchi & Liò, 2007). Due to the huge amount of data, these advances could have not been achieved without the support of bioinformatics algorithms.

Many tasks in bioinformatics are traditionally solved by search algorithms, such as BLAST (Altschul et al., 1990), which matches a sequence against a reference database containing annotated entries (and so tasks are solved in terms of similarity). Databases are usually very large and search can be very slow, impacting fields in which a timely response is important, such as cancer diagnostics. Indexing strategies or heuristic searches can speed up the querying process, at the expense of a space or accuracy trade-off.

The application of machine learning methods has provided a great contribution to this field, allowing to solve tasks in a computationally inexpensive way, with a greater robustness to noise, and to discover new associations (Vinga, 2014; Larranaga et al., 2006).

In particular, the recent development of Natural Language Processing (NLP) techniques has lead researchers to use advanced neural models for modeling biological sequences, such as proteins, DNA, and RNA (Ji et al., 2021; Zaheer et al., 2020; Wahab et al., 2021).

However, the heterogeneity in the databases where biological data are stored and the lack of a general consensus on the data processing makes it difficult to define machine learning tasks, and more in general to approach this domain without having a deep understanding of it. At the same time, there is a consensus in the NLP community that to properly evaluate a language model there is the

necessity to test in on multiple tasks to measure whether it can be broadly applied. For this reason, multi-tasks benchmarks such as SUPERGLUE (Wang et al., 2019a) have been developed. To meet these needs, Rao et al. (2019) have developed TAPE, a benchmark designed to rigorously evaluate the performance of learned protein embeddings through a multitude of tasks, encompassing the domains of structure prediction, evolutionary understanding, and protein engineering, which have similarities with NLP tasks such as named entity recognition and natural language classification.

A different class of languages, not covered by TAPE, ranges over the DNA and RNA domains. These domains are crucial in bioinformatics because they enable researching tumors, subcellular structures and the environment. Accordingly, we present BANANA (Benchmark for the Assessment of Neural Architectures for Nucleic Acids), a benchmark consisting of six classification tasks assessing language understanding performance in the DNA and RNA domains. The tasks are defined over three genomics (human DNA, bacterial 16S gene, nematoda ITS2 gene) and one transcriptomics (human mRNA) language and measure an architecture's capability of performing whole-sequence classification in a variety of setups. We curate existing data providing datasets that do not impose any specific constraint regarding the approach, creating a benchmark that is versatile and easy to use even for those who do not have knowledge of the domain. Finally, we perform few experiments with advanced neural architecture, so as to provide baselines for future works.

In Section 2 we describe existing benchmarks and draw similarities and differences with our work. Section 3 provides background for our domain of application and highlights the importance of our tasks, which are described in Section 4. We evaluate our baselines in Section 5, while Section 6 list related works for each task. Finally, Section 7 concludes.

## 2   NATURAL LANGUAGE UNDERSTANDING AND PREVIOUS BENCHMARKS

In the context of unstructured data, such as textual documents, their representation plays a key role in the performance of any machine learning algorithm. Indeed, representations capable of encoding information not only about the lexicon but also about the syntax and the semantics of a sentence, provide the models with additional knowledge that can be exploited in higher-level tasks. The task of learning effective representations is called Natural Language Modelling (NLM), and it is typically addressed through semi-supervised tasks (Devlin et al., 2019). Once learned, the quality of a language model can be evaluated by using it to address more advanced NLP tasks that concern the comprehension of the text, in a set of tasks that falls under the broad definition of Natural Language Understanding (NLU).

Our proposal draws inspiration from the GLUE benchmark (Wang et al., 2019b), a set of 9 supervised sentence understanding tasks in the English language based on existing datasets. Its authors proposed it with the purpose of stimulating the development of a more unified language model, capable of solving a range of different linguistic tasks in different domains. Their experimental results showed how models trained in multi-task learning setting outperformed models trained on a single task, and therefore the benefits of sharing knowledge across tasks. But the interest in most of these tasks has quickly decreased with the advent of advanced language models such as BERT, which was capable of easily solving most of them. Therefore, a new benchmark called SUPERGLUE (Wang et al., 2019a) has been later proposed, including new supervised learning tasks that are solvable by humans but are difficult for machines. All the tasks proposed in this benchmark regard the classification of one or multiple sequences of text, and a final score is given based on the average score obtained on every single task. Similarly, our tasks regard the classification of a single sequence, and we compute a final score in a similar fashion. Another analogy between these works and our proposal concerns the fact that they are agnostic regarding the pre-train of the architectures and do not include the language modeling task as part of the benchmark.

The TAPE benchmark (Rao et al., 2019) fills the need for datasets and standardized evaluation methods for machine learning techniques applied to protein sequences, and it is another source of inspiration for our work. The authors propose to use NLP techniques in the domains of proteins, presenting the self-supervised learning task of modeling the "languages of proteins" using a dataset containing proteins of different organisms and absolving different functions, along with 5 supervised learning tasks concerning 3 major subfields of protein sciences. All these tasks are based on existing datasets, but they are curated and formalized by Rao et al. The experimental part shows that the self-supervised task is beneficial for all the neural models and that they can successfully address three

of the five tasks. It also highlights how the performances of each model vary across the different sub-fields. Our work differs from TAPE since we address tasks that span across four different "biological languages" (human DNA, human mRNA, bacterial and archaebacterial DNA, nematoda DNA) and specifically address each of them. Instead, TAPE's tasks focus on multi-lingual settings, which requires the ability to generalize over a broad spectrum of "protein languages".

## 3 BIOLOGY BACKGROUND

The central dogma of molecular biology (Crick, 1970) states that, for every living organism, the information is stored in DNA (deoxyribonucleic acid), then it flows to RNA (ribonucleic acid) and finally to proteins, which are the building blocks of life but cannot be transferred from one generation to the next. Both DNA and RNA are composed of a sugar-phosphate scaffold and **nucleotides** (nt), which contain the genetic information. DNA is a double helix composed of two filaments in which complementary nucleotides (A, T, C, G) bind to each other forming base pairs (**bp**), while RNA is a single strand of nucleotides (A, U, C, G), capable of folding in different ways[1].

RNA is synthesized (*transcribed*) by making a complementary copy (using a different alphabet, but semantically a 1:1 copy) of a region of DNA (a gene), starting from the transcription start site (TSS) up to a termination point. In order for the copying machinery to "select" the proper region, some transcription factors (TF) must bind to DNA in a region immediately before the TSS, called the **promoter** of that gene. DNA is compressed into a structure called **chromatin**, whose folding structure plays an important role in cell differentiation: a liver cell and a neuron share the same DNA, but thanks to different chromatin profiles they appear and behave in completely different ways. These mechanisms are fundamental for life and their malfunction is linked with cancer (Morgan & Shilatifard, 2015).

RNA absolves many functions inside the cell (e.g., gene regulation, signaling, etc.) and it is also correlated to cancer (Reddy, 2015) and infections (Fani et al., 2021). Three main types play universal roles across every living organism: ribosomal RNA (rRNA) and transfer RNA (tRNA) are the gears in the machinery for protein synthesis, while messenger RNA (**mRNA**) constitutes the "recipe" for proteins. Freshly transcribed mRNA filaments migrate each to a specific position inside the cell, known as their **subcellular localization** (Holt & Bullock, 2009), and then ribosomes translate them to proteins. The analysis of mRNA localization may help to better understand subcellular compartments and lead to a more detailed and nuanced understanding of cellular architecture (Martin & Ephrussi, 2009).

Ribosomes translate mRNA into proteins and are made of two subunits: small (SSU) and large (LSU). The bacterial SSU is called **16S** and, since it evolves with a very slow mutation rate, it can be used to reliably classify bacteria, also for the so-called "microbial dark matter", those bacteria which cannot be classified using traditional methods (Kalmbach et al., 1997), and which can have an effect, for example, on health (Shreiner et al., 2015) or climate change (Jansson & Hofmockel, 2020). Eukaryotes can be classified by SSU markers (18S) as well, however popular alternatives are the Internal Transcribed Spacer 2 (**ITS2**), a region between the SSU and the LSU which is transcribed, but then destroyed instead of being part of ribosomes, and the mitochondrial SSU (12S).

*Sequencing* is the process of reading a sequence of DNA, RNA, or protein, starting from a sample. Genomics, transcriptomics, and proteomics are the studies of data (DNA, RNA, and proteins, respectively) sequenced from a single organism. When this is done on data sequenced from multiple organisms at the same time, the studies are called metagenomics, metatranscriptomics, and metaproteomics. For example, extracting all the DNA present in an environmental sample in order to determine which bacteria live in that environment is a classical example of metagenomics. Sequencing technologies are characterized by various kinds of reading errors, an important one arising during meta-omics sequencing is the production of **chimera** artifacts. Chimeras are digital sequences that do not exist in the real world, instead they are composed of portions of existing sequences that have been hybridized. They are produced when sequencing with highly parallel technologies is stalled while it is reading sequences that share a similar region. For example, if

---

[1]Since base pairs are complementary, from a linguistic perspective, a single DNA strand contains the entire information, so both bp, for DNA, and nt, for RNA, can be considered the "characters" of nucleic acids languages.

AACTCTGGA and GGGTCTTTT are both stopped and resumed when reading TCT, the sequencer may mistakenly read the chimeras AACTCTTTT and GGGTCTGGA. More complex chimeras may also arise as hybrids of more than two sequences, palindromic sequences, etc. Chimeras are a serious problem in metagenomics studies, especially on 16S genes, constituting even a large portion of sequencer outputs (Wang & Wang, 1997), and can cause misclassification of a population of organisms. Moreover, detection has been a problem and some public databases have been historically plagued by chimeras mistaken for legitimate sequences (Hugenholtz & Huber, 2003).

## 4 TASKS AND DATASETS

BANANA is a benchmark of 6 supervised learning bioinformatics tasks related to genomics. They have been chosen so they are similar in their formulation: the classification of a single sequence of RNA or DNA. Our aim is to provide a uniform input formulation, to make all the tasks compatible with any NLU model. At the same time, they have been selected to address a broad range of aspects and present slightly different characteristics, so that each of them provides a different challenge. Indeed, the tasks vary between binary and multi-class, single and multi-label, flat and hierarchical (Freitas & Carvalho, 2007), balanced and unbalanced distributions (Wald et al., 2013). Three additional criteria have been used for our selection: 1) the task must be solvable without explicit domain-related knowledge; 2) the data must not require excessive computational resources; 3) the data must be already available under a license that allows us to redistribute them.[2]

Even if our benchmark is based on existing data, such data is rather difficult to use in its original form as a benchmark, due to heterogeneity in the samples and labels across tasks, and because their use requires deep knowledge of the bioinformatics domain. Moreover, they do not include negative samples, a problem that previous authors have addressed in various ways, without a standardized methodology, as discussed in Section 6. We publicly release a curated version of those data, selected and formatted so as to make them approachable by anyone as machine learning tasks but without altering their nature nor their complexity. For each task, we define 3 data splits, allocating 75% of the data for training, 5% for validation, and 20% for testing.[3] Additional information regarding the original format of the data and their processing can be found in Appendix B.

It's important to note that, due to the nature of the tasks, and unlike traditional NLP, labeling cannot be done by human evaluation and requires years of (possibly expensive) lab experiments which are then merged together. To limit costs, it is not uncommon to gather data and then use traditional bioinformatics algorithms to label some or the entirety of it. These algorithms have been thoroughly validated by the scientific community, and therefore they can be considered a reliable gold standard.

For each task, we present a brief summary containing the main characteristics of the task, characteristics of the dataset, and which metric will be measured for the benchmark. We also include an analogy to natural language processing tasks, to facilitate understanding and to help to bridge the gap between the communities of bioinformatics and NLP. The metrics for each task range from 0 to 1 and we propose as the final BANANA score their arithmetic mean. We propose to use this score to quantitatively assess any neural architecture capable of solving these tasks, regardless of how it was initialized. This is not only to allow the evaluation to a broad spectrum of training processes (single task, self-supervised pre-training, contrastive learning, joint learning, etc.), but also to avoid penalizing families of architectures that could benefit from a given style (e.g. transformers pre-trained with language modeling tasks, or CNNs pre-trained as autoencoders).

### 4.1 PROMOTER DETECTION (*PromD*)

- **Type of task.** Binary classification on a balanced dataset.
- **Format of data.** Sequence of fixed length.
- **Language.** Human DNA. Alphabet: $\Sigma = \{A, T, C, G\}$.
- **Objective.** Establish whether a sequence of DNA contains a promoter or not.

---

[2]Appendix A contains examples of tasks we have considered but discarded.

[3] Following ICLR's FAQs, after the opening of the discussion forums, we will make a comment directed to the reviewers and area chairs and put a link to an anonymous repository. In case of acceptance, it will be hosted on the Zenodo website.

- **Dataset.** 59,194 sequences of length 2048 associated with a single binary label.
- **Main metric.** Macro average F1 score.
- **NLP analogy.** Positive/negative sentiment analysis.

## 4.2 CHROMATIN CLASSIFICATION (*ChromC*)

- **Type of task.** Multi-label classifications with many classes.
- **Format of data.** Sequences of variable length.
- **Language.** Human DNA. Alphabet: $\Sigma = \{A, T, C, G\}$.
- **Objective.** Determine all the possible ways a DNA strand can fold.
- **Dataset.** 4,862,738 sequences of length of at most 1000bp. Each of them is associated with 919 binary labels.
- **Main metric.** Area under ROC curve (AUC).
- **NLP analogy.** Joint classification of multiple related concepts (eg. legal document analysis according to obligations, penalties, and clarity).

## 4.3 MRNA CLASSIFICATION (*mRNAC*)

- **Type of task.** Multi-label classifications with few classes.
- **Format of data.** Sequences of variable length.
- **Language.** Human mRNA. Alphabet: $\Sigma = \{A, U, C, G\}$.
- **Objective.** Establish from which subcellular location the sequence comes from; in case of multiple possible source, multiple labels are assigned.
- **Dataset.** 127,208 sequences of length ranging from 33nt to 9,256nt (median length about 4400nt). Each sample is associated with 15 binary labels.
- **Main metric.** AUC.
- **NLP analogy.** Multiple topic classification.

## 4.4 16S HIERARCHICAL CLASSIFICATION (*16sH*)

- **Type of task.** Multi-class hierarchical classification (7 levels) with many samples.
- **Type of data.** Sequences of variable length.
- **Language.** Bacterial and archaebacterial DNA. Alphabet: $\Sigma = \{A, T, C, G\}$.
- **Objective.** Provide a complete classification (kingdom, phylum, class, family, order, genus, species) for a bacteria from its 16S gene.
- **Dataset.** 1,783,227 sequences of length between 890 and 4000bp. Each of them is associated with 26,793 categorical labels organized into 7 levels.
- **Main metric.** The average score between the 7 macro-averaged F1 scores computed for each level of the hierarchy.
- **NLP analogy.** Hierarchical language identification on a large dataset (eg. Indoeuropean > Germanic > North Sea Germanic > English vs. Indoeuropean > Romance > Italo-Dalmatian > Italian).

## 4.5 ITS2 HIERARCHICAL CLASSIFICATION (*its2H*)

- **Type of task.** Multi-class hierarchical classification (4 levels) with few samples.
- **Type of datas** Sequences of variable length.
- **Language.** Nematoda (round worms) DNA (ITS2 gene). Alphabet: $\Sigma = \{A, T, C, G\}$.
- **Objective.** Provide a partial classification (class, family, order, genus) for a nematoda from its ITS2 gene.
- **Dataset.** 19,222 sequences of length betweeen 100bp and 400bp . Each is associated with 277 categorical labels organized in 4 levels.

- **Main metric.** The average score between the 4 macro-averaged F1 scores computed for each level of the hierarchy.
- **NLP analogy.** Hierarchical language identification on a small dataset.

## 4.6 CHIMERA DETECTION (*ChimD*)

- **Type of task.** Binary classification with an unbalanced dataset.
- **Type of data.** Sequences of variable length
- **Language.** Bacterial and archebacterial DNA (16S gene). Alphabet: $\Sigma = \{A, T, C, G\}$.
- **Objective.** Determine if an input is a real bacterial sample or a chimeric artifact.
- **Dataset.** 412,251 sequences, 19,631 of which labelled as positive. The length of each sample ranges from 1,111bp to 2,435bp.
- **Main metric.** Macro average F1 score.
- **NLP analogy.** Semantic coherence from grammatically correct samples. E.g., "The dog jumped over the fence in order to fetch the stick" and "Transformers can be pre-trained in order to achieve state-of-the-art performance" are two coherent sentences, but their chimera "The dog jumped over the fence in order to achieve state-of-the-art performance" is not, in spite of still preserving a well formed structure thanks to the shared portions.

## 5 EXPERIMENTAL SETUP

As baselines, we evaluate two neural architectures, with different pre-training configurations. Since our benchmark includes only original datasets, it is not possible to compare these approaches to previous state-of-the-art results. In our experiments, we focused on models capable of extracting token-level contextual embeddings which are used to feed a task-specific classification head, as proposed by Devlin et al. (2019). However, we want to underline that even if we have created this benchmark primarily for the evaluation of contextual embedding-based architectures, it can be used for any type of classifier.

### 5.1 ENCODING

Unlike natural languages, DNA and RNA sequences are not "naturally" split into words, therefore it is necessary to define synthetic tokens. A popular approach in bioinformatics is to use $k$-mers, which are "words" made of $k$ contiguous characters (Liang, 2012). We perform input tokenization with a two-step approach: first, we split the input into $k$-mers, then we apply byte-pair encoding (Sennrich et al., 2016) to compress the embedding table and avoid out-of-vocabulary tokens. In each of our experiments, we set $k = 9$ and the size of vocabulary at 32,000. These values were identified as a good compromise between performance and input size on preliminary tests.

### 5.2 MODELS

We used LongFormer (Beltagy et al., 2020), a sparse-attention transformer, as "state-of-the-art" baseline for producing contextual embeddings. For each task we test three training approaches: 1) a single-language single-step training, 2) a single-language two-steps training, 3) a multi-lingual two-steps training. The first approach is to directly train a different LongFormer model on each task (LONGFORMER-NO). The second one differs because we introduce a pre-training step for each model. Firstly we do a self-supervised dynamic masked language model pre-training (Liu et al., 2019), then a supervised fine-tuning on the classification task using a smaller learning rate (Devlin et al., 2019). Both steps are performed using only the data provided for that specific task, therefore we address this approach as LONGFORMER-MONO. The last approach consists of pre-training a single LongFormer, shared across all the tasks, and then fine-tuning a different copy of it on each of the 6 tasks independently. The pre-training is done on a "multilingual" dataset created by randomly sampling 200,000 sequences from each of the six tasks, using oversampling techniques for small datasets. To keep a similar computational budget to the mono-task case, the embedding size of the LONGFORMER-MULTI is reduced from 256 to 192 dimensions, and the number of attention heads

Table 1: Models' performance on the benchmark test sets, reported as percentage points (LF=LONGFORMER).

| Model | PromD | ChromC | mRNAC | 16sH | its2H | ChimD | BANANA |
|---|---|---|---|---|---|---|---|
| | *F1* | *AUC* | *AUC* | *F1* | *F1* | *F1* | *AVG* |
| LF-NO | 46.3 | 73.8 | 70.0 | **76.4** | 72.2 | 86.0 | 70.8 |
| LF-MONO | 74.0 | **79.1** | **84.5** | 76.3 | 71.2 | **87.6** | **78.8** |
| LF-MULTI | **77.7** | 51.7 | 66.4 | 69.0 | 56.7 | 79.2 | 66.8 |
| BILSTM-NO | 61.1 | 51.1 | 80.4 | 73.8 | 60.2 | 78.8 | 67.6 |
| BILSTM-MULTI | 73.2 | 51.2 | 82.2 | 70.3 | **74.5** | 72.3 | 70.6 |
| RANDOM | 50.0 | 50.0 | 50.0 | 3.5 | 7.5 | 50.0 | 35.1 |
| MAJORITY | 33.3 | 50.0 | 50.0 | 8.0 | 16.9 | 33.3 | 31.9 |

in the backbone from 16 to 12. The other hyperparameters are kept the same as the mono-task case. Additional details are provided in Appendix C.

As a "traditional" baseline, we trained a three-layer bidirectional LSTM, the hidden state of the last layer constitutes the contextual embeddings provided to the classification heads. Two versions of this baseline were trained separately: one directly on the final task (BILSTM-NO) and the other in two stages, following the multilingual setting (BILSTM-MULTI). For comparison, we also provide two deterministic lower-bound baselines: random classification and majority class.

In all cases, the pre-train lasted for a maximum of 10 epochs, with early stopping either when 24 hours elapsed or when the perplexity on the validation set dropped below 5. For fine-tuning, we trained for a maximum of 10 epochs, with early stopping in case of overfitting on the training set (measured as a sharp drop of performance on validation in the last epoch) or up to 12 hours of training.

## 5.3 SUPERVISED TASKS TRAINING

For the binary classification tasks (**PromD** and **ChimD**) the contextual embeddings are subject to a 0.1 dropout and then fed to a fully-connected single neuron with sigmoid activation. We use *binary cross-entropy* loss for **PromD** and the weighted version and for **ChimD** to account for class imbalance, giving a weight of 0.95 to the positive class and 0.05 to the negative class.

We use a similar setting for multi-label classification tasks (**ChromC** and **mRNAC**), but we use a different sigmoid layer for each label. The networks are trained using the *binary cross-entropy loss* averaged across labels. In **ChromC** task, we address the classes unbalance following Zaheer et al. (2020), assigning a weight of 8 to the positive class and 1 to the negative one because they have empirically proven to improve convergence with respect to the unweighted binary cross-entropy.

To address the hierarchical classification tasks (**16sH** and **its2H**) we exploit hierarchical neural attention (Galassi et al., 2020) as shown in Figure 1. First, we compress contextual embeddings to a lower-dimensional space through a 1D convolutional layer, to meet our limitations in terms of hardware. Then, the 32-dimensional embeddings are used both as keys and values of a multi-head attention layer (Vaswani et al., 2017) for each hierarchy level. The query element for the top level of the hierarchy (*kingdom* for **16sH** and *class* for **its2H**) is the same vector used as keys and values, while for every other hierarchy level, the query element is the output of the previous attention layer. The outputs of each attention module (one for each hierarchy level) are input to a fully-connected layer with softmax activation which constitutes the output classification for that level. The labels are one-hot encoded and each level of the hierarchy is trained on the average *categorical cross-entropy loss* across all layers.

## 5.4 RESULTS

Both the two neural architectures greatly surpass the baselines in almost all the tasks. The **ChromC** task seems to be the more challenging one, with only the LONGFORMER-NO and -MONO significantly overcoming the baseline of 50%. **ChimD** results to be the easiest to solve, with all the

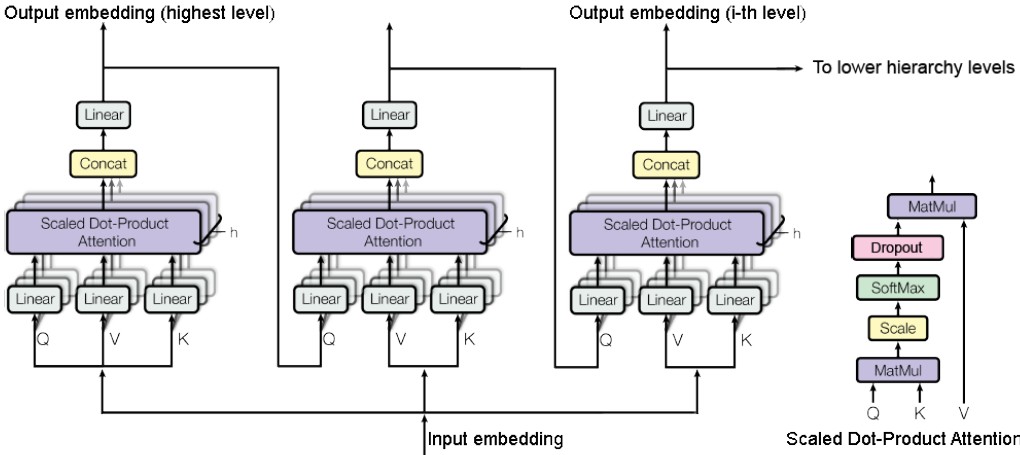

Figure 1: Hierarchical multi-head attention for **16sH** and **its2H** classification heads, partially based on Vaswani et al. (2017).

models achieving more than 70 percentage points, up to 87%. The LONGFORMER-MONO is the model that obtains the best final score, and it also achieves satisfactory results in all the tasks. The LONGFORMER-MULTI outperform the MONO version only on **PromD**, and is less performant than the other LONGFORMERS on all the other tasks. This may be caused by the fact that a multi-lingual representation is less specialized, but it may also be influenced by the fact that we use a smaller number of attention heads and the fact that the MULTI pre-train is done only on a portion of the data used for the MONO pre-train. On the contrary, BiLSTM-MULTI and BiLSTM-NO have a comparable behavior in most of the tasks, with a pronounced difference only on **its2H**, where the MULTI approach achieves the better score.

None of the models completely dominates any other, and different tasks are solved better by different models. These heterogeneous performances confirm the claims of Rao et al. (2019): a single task is not enough to properly evaluate a trained model and there is a need for multi-task benchmarks such as TAPE and BANANA.

Other than the formal BANANA score, we strongly encourage considering also the environmental impact of the architectures. Indeed, as measured by Strubell et al. (2019), the development of state-of-the-art solutions often has a massive carbon footprint. With the perspective of a sustainable research, it is important to shift toward a Green AI focused on efficiency, not only accuracy. Small-size architectures are therefore preferable, since the number of parameters can influence the time, and therefore the energy consumption, needed to train a model (Schwartz et al., 2020). We include details about the size of our models in Appendix C.

## 6    RELATED WORKS

Umarov & Solovyev (2017) were the first to train a neural network on promoter detection. They solved the task by using a very small CNN on a limited subset of promoters. They created a dataset of negative samples by choosing coding sequences (i.e., portions always ahead of promoters). The ease with which the task was solved has been contested both by Qian et al. (2018) and Oubounyt et al. (2019) since it contains "useful" biases which can help a classifier, like keeping the TSS of the positive classes always in the same position. Oubounyt et al. have proposed to create negative samples by means of random noise, but we advocate that, from a language modeling perspective, this approach is unsound. Indeed, it shifts the task of recognizing a well-formed language (the promoter's) to a much simpler discrimination task between a coherent sequence of syntagms and an incoherent sequence of symbols. Zaheer et al. (2020) were the first to solve the task pre-training a Transformer-based architecture, BigBird, on external data (the entire human genome, version hg19), achieving near-perfect performance. However, their approach to negative samples is the same as

Oubounyt et al.. Ji et al. (2021) addressed the task with another Transformer-based architecture and synthesizing two different types of negatives, but without shifting the TSS in the positive samples.

The chromatin classification task has been introduced by Zhou & Troyanskaya (2015) and has gained huge traction in the bioinformatics community. They propose to approximate the continuous domain of possible foldings into a finite number of classes and to address the problem as a multi-label classification task. We follow such an approximation, that allows us to avoid two main problems with the general task of chromatin folding. The first one is that predicting the actual shape requires a deep knowledge of the domain and possibly very complex architectures, akin to protein folding (Jumper et al., 2021). Additionally, the same sequence, even from the same cell, may change its shape over time, so unlike protein folding, there would be more than one output, possibly of an unknown a priori number. Differently from our benchmark, Zhou & Troyanskaya propose a test set that contains only sequences from human chromosomes 8 and 9, possibly biasing the evaluation scores in different ways in the two cases. Zaheer et al. (2020) address the task achieving very good results, but are not directly comparable to ours due to the different data splits.

The tasks belonging to the family of subcellular localization have been solved in many different flavors using various architectures and many different datasets, addressing protein (Sønderby et al., 2015), mRNA (Yan et al., 2019; Wang et al., 2021; Li et al., 2021), mitochondrial proteins (Savojardo et al., 2020), etc. This heterogeneity testifies the practical importance of these tasks but makes comparison among architectures impossible. Our proposal of mRNA localization aims to cover this limitation by providing a well-defined prototypical task for the entire family.

Marker gene metagenomics (16S and ITS2 on fungal sequences) is an area of active research, also because meta-analyses have revealed the presence of classification biases in traditional pipelines (Nearing et al., 2018; Straub et al., 2020). Neural approaches to metagenomics range from traditional architectures to ad-hoc ones: Fiannaca et al. (2018) use CNNs and deep belief networks, while Fioravanti et al. (2018) design Ph-CNN introducing a new convolutional layer operating on multiple input sequences aggregated by k-nearest neighbors. It is important to note that, like other neural approaches, our hierarchical classification tasks play only a small role in a complete metagenomics setup. Indeed, it requires to classify millions of sequences in presence of noise, chimeras, and other artifacts, so it typically requires domain knowledge and ad-hoc denoising algorithms, e.g., DADA2 (Callahan et al., 2016).

To the best of our knowledge, we are the first to propose the use of neural architectures for the classification of Nematoda ITS2 sequences (which is basically a smaller marker-gene metagenomics task) and chimera detection tasks. These tasks have been solved with well-established algorithms in the bioinformatics community.

Regarding chimera detection, traditional algorithms, like Chimera Slayer (Haas et al., 2011), exploit a database of chimera-free sequences by counting how many results are triggered by the query sequence. UCHIME (Edgar et al., 2011) improves performance by computing the reference database from the sample itself, under the assumption that, in a metagenomic sample, the abundance of legitimate sequences is higher than the one of chimeras.

## 7    CONCLUSION

We presented BANANA, a new benchmark to assess the capability of neural models to understand biological languages. The purpose of this benchmark is to narrow the gap between the communities of bioinformatics and NLP (and machine learning in general), providing a standardized and reliable tool to evaluate state-of-the-art neural techniques on bioinformatics tasks. Our benchmark is based on existing data, which have been curated and formatted into 6 datasets for which we define as many supervised learning tasks. These datasets are easy to use and can be addressed by machine learning and NLP experts without requiring any additional knowledge of the domain. Our experiments with two neural baselines confirm the results of Rao et al. (2019), remarking the usefulness of multiple-tasks benchmark to properly evaluate neural architectures.

In future works, we want to investigate multi-lingual pre-training further, with the purpose to develop a single neural model that can address successfully all the tasks. We also want to pursue more efficient models, and to extend the BANANA score by defining with formal Green AI metrics, capable of assessing environmental impact and efficiency.

ETHICS STATEMENT

Our work is compliant with the ICLR Code of Ethics. Our benchmark includes only data that are already publicly available under licenses that allow us to redistribute them. We report the sources of our data, describe their original format, and explain our formatting procedure in Appendix B.

REPRODUCIBILITY STATEMENT

The dataset, as well as the code used to run the experiments, and the code used to process the data will be released publicly in case of acceptance. The dataset will be made publicly available and hosted on the Zenodo website, the code will be put on a publicly available GitHub repository. During the review period all the material will be available to the reviewers. Following ICLR's FAQs, after the opening of the discussion forums, we will make a comment directed to the reviewers and area chairs and put a link to an anonymous repository.

Any details regarding our models and training hyper-parameters are specified in the body of the paper or the Appendices.

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

Table 2: Dataset composition for each task.

| Split | PromD | ChromC | mRNAC | 16sH | its2H | ChimD |
|---|---|---|---|---|---|---|
| **Train** | 44,394 | 3,647,053 | 95,406 | 1,337,420 | 14,416 | 402,435 |
| **Validation** | 2,959 | 243,136 | 6,360 | 89,161 | 961 | 1,963 |
| **Test** | 11,839 | 972,549 | 25,442 | 356,646 | 3,845 | 7,853 |
| **Total** | 59,192 | 4,862,738 | 127,208 | 1,783,227 | 19,222 | 412,251 |

## A    DISCARDED TASKS

While designing this benchmark, we decided to focus on classification tasks that did not require, character-level reasoning, domain knowledge, or complex architectures (note that we decided to use the neural attention mechanism for our baselines in the hierarchical tasks, but this was not imposed by the nature of the tasks). For these reasons, we ruled out every folding task, including chromatin prediction in terms of spatial conformations or graph representations, RNA structure prediction (Piekna-Przybylska et al., 2007), protein folding (Jumper et al., 2021), etc.; every regression task, such as TSS localization inside a promoter, molecular clock regression (Hasegawa et al., 1984), a full marker-gene metagenomic pipeline (which also has the complications of domain knowledge and the need of processing extremely large inputs), etc.; every alignment task (given a set of noisy fragments and a reference database, sort and merge the fragments in order to get a sequence as close as possible to one in the database), including shotgun metagenomics; every sequence-to-sequence task, such as post-translational RNA modifications and intron splicing (Piva et al., 2012); and every multiple-sequence classification task, like virus-host RNA interactions (Li et al., 2015) and chromatin-chromatin interactions (Xie et al., 2016).

Another important factor in our decisions was the desire to provide a set of tasks as varied as possible, while still keeping enough correlation to allow multilingual approaches to the benchmark. For this reason, we discarded all the protein-related tasks, since linguistic differences arise even at the alphabet level (DNA and RNA have 4 symbols, while proteins 22). Moreover, protein tasks tend to require a deeper degree of domain knowledge and often rely on additional features other than the sequence.

Finally, we discarded tasks that, albeit interesting on a biological level, had either too few samples or were too similar to tasks already present in the dataset. For example, 12S mitochondrial metagenomics (Iwasaki et al., 2013) has only singleton labels, ITS2 fungal metagenomics (Abarenkov et al., 2010) is too similar to other tasks (its sequences are as short as our **its2H** and almost as many labels as our **16sH**), other variants of subcellular localization, are too similar to **mRNAC**.

## B    SOURCE OF DATA AND DATA PROCESSING

All the data we have included in our benchmark are derived from existing databases and, in most cases, have been processed to guarantee structured, fair, and challenging tasks. hereby report the original source of our data, describe in detail their original format, we list all the processing steps we have applied. This information is valuable both because guarantees that our work is reproducible, but also guarantees the absence of biases or other synthetic artifacts in our benchmark. All the code for data processing is publicly available.[4] Table 2 reports detailed statistics regarding the final size of our datasets.

PROMOTER DETECTION (*PromD*)

- **Source of data.** Human promoters from Eukaryotic Promoter Database (EPD):[5] 29,598 sequences of human promoters, 16,000bp long, and with the transcription start site (TSS)

---

[4]Following ICLR's FAQs, after the opening of the discussion forums, we will make a comment directed to the reviewers and area chairs and put a link to an anonymous repository. The code will be put on a GitHub repository in case of acceptance.

[5]https://epd.epfl.ch/index.php.

centered at position 10,000. Each promoter can contain one or more of the TATA box, Initiator motif, CCAAT box or GC box patterns.

- **Data Processing.** From each sequence we extract two subsequences of length 2048bp to create one positive and one negative sample. The positive sample is created by using a 2048bp window centered at the TSS and shifting it by a random offset, but guaranteeing that the TSS is inside the window and has at least a 32bp context on both sides. The negative sample is taken randomly placing the window before or after the positive one, maintaining a distance of at least 2000bp between the two.

CHROMATIN CLASSIFICATION (*ChromC*)

- **Original data.** Princeton University's DeepSea[6] training bundle (Zhou & Troyanskaya, 2015): 4,862,738 sequences of length 1000bp. Each sequence is associated with 919 binary labels which can be grouped into 3 categories according to the biological domain (DNAse I hypersensitivity sites: 125, Transcription factor binding sites: 690, Histone methylation sites: 104). Labels are computed by splitting the human reference genome (version hg19[7]) into 200bp bins, discarding the ones not associated with transcription factor binding events and classifying them according to the ENCODE (Sloan et al., 2016) and Roadmap Epigenomics (Chadwick, 2012) databases. Each sample is extended to 1000bp by taking the two 400bp regions adjacent to the classified bin.

- **Data Processing.** The human reference genome contains some small gaps of unknown content, marked as N characters in the sequences. We remove from each sample these gaps, making the sequences of variable length. In order to conform to the other tasks, we also reshuffle the original dataset and create new train/validation/test splits.

MRNA CLASSIFICATION (*mRNAC*)

- **Original source of data.** RNALocate 2.0.[8] More than 210,000 RNA-associated subcellular localization entries including messenger RNA (mRNA), micro RNA (miRNA) and other non-coding RNA (ncRNA) of various species.

- **Data Processing.** We extracted the human mRNA from the database and kept the annotations only for the 15 labels which were represented by at least 50 samples.

16S HIERARCHICAL CLASSIFICATION (*16sH*)

- **Original source of data.** Silva 138.1.[9] About 12 millions quality-checked ribosomal RNA (rRNA) sequences for all the three domains of life (Archaea, Bacteria, and Eukarya). Sequences are divided into small and large ribosomal subunits and are associated with their taxonomical name, up to the lowest level for which classification is possible. Unknown labels don't follow a uniform naming system.

- **Data Processing.** We extracted only the 16S genes (Archaea and Bacteria) from the small subunit database by filtering based on domain names and then retro-transcribed each sequence from RNA to DNA[10]. Each name is processed in order to have a single nomenclature for unidentified taxa. We encode each label with a integer number, using 0 for the *unidentified* class, and following the alphabetical order for the others. If level $i$ is labeled as *unidentified*, all the following levels in the hierarchy $i + 1, i + 2, \ldots, i + n$ are labeled in the same way, to guarantee a full hierarchy for each sample. We exclude any class for which only one sample is available. Splits are made so to guarantee that both the training and test set have at least one sample for each label.

---

[6]http://deepsea.princeton.edu/help.

[7]https://www.ncbi.nlm.nih.gov/assembly/GCF_000001405.13/.

[8]http://www.rna-society.org/rnalocate.

[9]https://www.arb-silva.de/no_cache/download/archive/release_138_1

[10]From a fingerprinting perspective, ribosomal markers (16S, ITS2, etc.) can be read indifferently as DNA or RNA, however RNA analysis typically focuses on analyzing mRNA, treating rRNA and tRNA as contaminants (Li et al., 2013), so in practice, ribosomal markers are analyzed as DNA sequences.

Table 3: Hardware specifications.

| Architecture | CPU | GPU | RAM | Swap |
|---|---|---|---|---|
| Architecture 1 | Intel i7-11370H | 8Gb nVidia RTX 3070 | 40 Gb | 64 Gb |
| Architecture 2 | AMD Ryzen 7 4800H | 6Gb nVidia RTX 2060 Max-Q | 16 Gb | 32 Gb |

Table 4: Generic hyperparameters for our experiments (N: no pretraining, MO: mono-task pretraining, MU: multilingual pretraining).

| Hyperparameter | Value |
|---|---|
| **LongFormer** | |
| MLM learning rate | $10^{-4}$ |
| Final task learning rate | $5 \cdot 10^{-5}$ |
| Embedding size (N/MO) | 256 |
| Embedding size (MU) | 192 |
| Hidden layers | 2 |
| Attention heads (N/MO) | 16 |
| Attention heads (MU) | 12 |
| Attention window | 32 |
| Attention dropout | 0.1 |
| **Bidirectional LSTM** | |
| Embedding size | 256 |
| Nr. hidden units | 256 |
| Hidden layers (BL) | 3 |
| Dropout between layers | 0.1 |

ITS2 HIERARCHICAL CLASSIFICATION (*its2H*)

- **Original source of data.** Nemabiome.[11] 19,300 sequences of ITS2 genes sampled from larval stages of parasitic worms. Each sequence is associated with a taxonomic name determined morphologically by inspecting each larva under the microscope.

- **Data Processing.** Same as *16sH*, but the higher taxa (domain, kingdom, phylum) are discarded because they are trivial (unary classification) and the species level is removed since it causes the majority of labels to have only one sample.

CHIMERA DETECTION (*ChimD*)

- **Original source of data.** GreenGenes (DeSantis et al., 2006) discarded chimeras and clean samples.[12] At each revision, the database is scanned for chimeras and they are removed. Most of the chimeric sequences are still accessible for manual download at NCBI using a list of chimeric IDs (21,169). Non-chimeric sequences are 1,262,986 bacterial 16S samples.

- **Data Processing.** We downloaded the chimeras from NCBI assigning a positive label. For each of them, we randomly sampled 20 bacterial sequences from GreenGenes. Validation and test set splits have a balanced (1:1) ratio of positive and negative samples, while the training set contains a 20:1 negative to positive ratio.

## C HYPER-PARAMETERS AND HARDWARE ARCHITECTURE

We trained the BiLSTM and LongFormer models on two hardware architectures, specified in Table 3. Tables 4 and 5 report the hyper-parameters used for the design and the training of our models.

---

[11]https://www.nemabiome.ca.
[12]https://greengenes.secondgenome.com/?prefix=downloads/greengenes_ database.

Table 5: Task-specific hyperparameters for our experiments.

| Hyperparameter | PromD | ChromC | mRNAC | 16sH | its2H | ChimD | Multilingual |
|---|---|---|---|---|---|---|---|
| MLM epochs | 10 | 6 | 3 | 3 | 3 | 3 | 2 |
| Epochs | 10 | 2 | 5 | 3 | 5 | 3 | (Same) |
| Input tokens | 512 | 128 | 512 | 256 | 256 | 256 | 512 |

Table 6: Trainable parameters of our models.

| | PromD | ChromC | mRNAC | 16sH | its2H | ChimD |
|---|---|---|---|---|---|---|
| **No Pretraining, Mono-task Pretraining** | | | | | | |
| **BiLSTM** | | | | | | |
| Backbone | 12,392,448 | 12,392,448 | 9,478,400 | 12,392,448 | 7,355,904 | 12,392,448 |
| Head | 262,145 | 60,228,503 | 3,932,175 | 30,709,437 | 3,719,423 | 131,073 |
| Total | 12,654,593 | 72,620,951 | 13,410,575 | 43,101,885 | 11,075,327 | 12,523,521 |
| **LongFormer** | | | | | | |
| Backbone | 12,465,408 | 12,367,104 | 9,551,360 | 12,399,872 | 7,363,328 | 12,399,872 |
| Head | 131,073 | 30,114,711 | 1,966,095 | 30,701,245 | 1,859,839 | 65,537 |
| Total | 12,596,481 | 42,481,815 | 11,517,455 | 43,101,117 | 9,223,167 | 12,465,409 |
| **Multi-lingual Pretraining** | | | | | | |
| **BiLSTM** | | | | | | |
| Backbone | 12,392,448 | 12,392,448 | 12,392,448 | 12,392,448 | 12,392,448 | 12,392,448 |
| Head | 262,145 | 240,911,255 | 3,932,175 | 61,398,717 | 3,719,423 | 262,145 |
| Total | 12,654,593 | 253,303,703 | 16,324,623 | 73,791,165 | 16,111,871 | 12,654,593 |
| **LongFormer** | | | | | | |
| Backbone | 14,503,168 | 14,503,168 | 14,503,168 | 14,503,168 | 14,503,168 | 14,503,168 |
| Head | 131,073 | 120,456,087 | 1,966,095 | 59,512,509 | 2,638,079 | 131,073 |
| Total | 14,634,241 | 134,959,255 | 16,469,263 | 74,015,677 | 17,141,247 | 14,634,241 |

Due to our limited resources, each model was trained with a maximum batch size of 32 samples, reduced (up to 4 for the multilingual BiLSTM trained on the **16sH** task) in case of out-of-memory errors. Both dynamic masked language modeling and final task training were subject to a limit of 10 epochs, reduced in case of overfitting or a timeout. We chose the input size as the smallest power of 2 capable of holding the majority of inputs since BPE encoding makes the input size variable. For tasks with greatly varying lengths, such as **mRNAC**, we decided to truncate inputs longer than the median value. Multilingual models used the maximum available input size (512 tokens). Table 5 summarizes the task-specific hyper-parameters used for training. The embeddings for tasks **16sH** and **its2H** are reduced to a 32-dimensional vector, using a 1D convolutional layer, with Gaussian error linear unit activation ($kernel\_size = 1, output\_channels = 32$). The hierarchical attention layer shown in Figure 1 is instantiated with a dropout rate of 0.1 and 8 attention heads for each level of the hierarchy. The sizes of each architecture, measured as the number of its trainable parameters, are listed in Table 6.

