# OpenReview forum: "BANANA: a Benchmark for the Assessment of Neural Architectures for Nucleic Acids"
_ICLR.cc/2022/Conference — ICLR 2022 Submitted_

### Official Review · Reviewer_xYqo · 2021-11-01

**Correctness:** 3
**Technical Novelty And Significance:** 2
**Empirical Novelty And Significance:** 3
**Recommendation:** 5
**Confidence:** 3

**Main Review:**

I am not particularly familiar with these classification tasks so I can not speak to their coverage of relevant tasks in the field. From my vantage point, this paper has the following strengths:

1) It provides a benchmark for an interesting problem area that is actively growing.
2) The tasks are standardized to allow greater reproducibility and comparisons across competing methods. Based on the authors descriptions, this appears to be missing in the field.
3) The authors provide reasonable baselines.

The paper has the following weaknesses

1) Unlike TAPE, which deals exclusively with protein sequences, this work proposes benchmarking DNA and RNA-based prediction. This appears to be convolving two different biological tasks (DNA and RNA) that I would think should be treated separately. Even within DNA and RNA there seem to be differences in, for example, human v.s. nematoda vs bacteria vs achaebacteria that complicate whether a single model should really be benchmarked across all of these tasks. Furthermore, the authors state they draw inspiration from GLUE / SuperGLUE which all deal with a single language (English). In summary TAPE and GLUE / SuperGLUE provide benchmarks capable to benchmarking a unified language model across different tasks. This does not feel appropriate for the tasks provided in this paper as there is no single language. Additionally, whereas in NLP one might hope to combine languages, here DNA and RNA represent fundamentally different domains and a better analogy seems to be benchmarking a model on an NLP task (DNA) and a computer vision task (RNA). In summary, while the goal of providing standardized datasets and benchmarks is achieved, the paper seems fundamentally flawed in it's desire to provide a GLUE / TAPE-like benchmarking system capable of evaluating self-supervised embeddings across different tasks.

In order to fix this the authors should address the following:
1a) Clarify their use of the term "multi-lingual" which appears to be incorrectly attributed to TAPE. Protein sequences are one language and TAPE provides many tasks for that one language just as GLUE provides many tasks for the English language.
1b) Comment on the mixing of multiple domains into one benchmark and compare / contrast with SuperGLUE and TAPE. Explicitly discuss the issue of a universal language model, which seems inappropriate for this task. Comment on the results of LF-MULTI and BiLISTM-MULTI in this context. Given the issues I pointed out above, it seems unsurprising that these approaches tend to perform worse (e.g. LF-MULTI is worse on average than LF-NO and LF-MONO).
1c) Currently, this seems to me to be a major issue with this benchmark. I would greatly appreciate any discussion that the authors can provide to help me better understand whether my critiques are fair and how they can or cannot be addressed.

My remaining critiques are smaller:
2) The "Objective" for 4.2 Chromatin Classification seems vague and I could not understand what the task is or how this becomes a classification task. Perhaps this could be made clear without resorting to the appendix?

3) The section "Encoding" was a bit vague. Why are k-mers needed at all instead of just using bases as tokens analogous to how amino acids are used as tokens in TAPE. Furthermore there is no comment on whether there is a universal encoding for both RNA and DNA or whether the tokenization is done separately for each. It appears to be the former, otherwise the MULTI pretraining would not make sense, but this should be more clearly explained and justified in the paper.

**Summary Of The Paper:**

This paper presents a benchmark consisting of six classification tasks that make predictions based on DNA or RNA sequences. The benchmark is an attempt to standardize existing datasets in terms of featurization, train/val/test splits, and performance metrics. It also reduces the amount of domain expertise required to benchmark a model on these tasks. The paper provides two neural network architecture baselines as well as different pre-training strategies and finds that no single model/strategy dominates all of the tasks.

**Summary Of The Review:**

As the paper currently stands, I see major issues with this being a useful benchmark for the community. It appears to be mixing too many domains (e.g. RNA vs DNA, human vs bacteria). I am open to changing my mind after discussing with other reviewers who may know more about this field and after hearing feedback from the authors. I thank the authors for tackling this challenging and often overlooked problem of benchmarking, but I would like to see more care taken in the construction of the benchmark before this paper is ready to be accepted.

---

> ### Author Response · Authors · 2021-11-19
> **.**
>
> We thank the reviewer for the helpful review. Since some reviewers asked us to run the experiments in a different setting, we are not able to submit a revised version of the manuscript in time. Nonetheless, to help us improve our work, we would appreciate it if the reviewer would take the time to read our rebuttal and comment on our answers.
>
> We find the discussion about the definition of “language” in this context very interesting and stimulating and we would greatly appreciate additional inputs from the reviewer on this topic. In the following paragraphs, we will explain our reasoning in more detail.
>
> Formal definitions of “language” are clearly not applicable in the context of genomics. In our work, we have used an approximation of the notion of language in an effort to find similarities between the Bio and the NLP communities. Neural networks-based models rely on statistical relations across sequences rather than proper linguistic concepts, and therefore we think they can be applied in both domains in a similar manner.
>
> An informal definition in this context can be given by the criterion used to establish whether two sequences belong to the same language or not, based on how we want to frame a particular problem.
> We consider sequences coming from two different species belonging to different languages. With this boundary, TAPE’s pretraining corpus (Pfam) is multi-lingual, since it is defined over many different species, ranging from viruses to superior Eukaryotes, and so is their Remote Homology Detection task. This definition is, in our opinion, further strengthen by the fact that homologies can be better framed in a (linguistic) phylogeny task, because natural evolution and linguistic evolution share useful “features” which can be exploited to solve the task (eg. prefixes shared across two related languages), while a mono-lingual frame would consider two homologs as synonyms and call for different kinds of “features” (eg. in which context the two “words” appear more often), which are less useful for that task, in our opinion.
>
> A, still valid, mono-lingual framing for TAPE would be to state that every protein belongs to the same alphabet of amino acids, and their chemical interactions are uniquely determined by the sequence itself, the medium conditions (eg. pH), and nearby molecules. This is certainly a useful framing as well, and in fact, this is more useful than a multi-lingual one for their Secondary Structure Prediction task. Is this the kind of framing the reviewer had in mind when stating TAPE is mono-lingual?
>
> Under the objection that DNA and RNA can be considered not only different languages, but entirely different fields (like CV and NLP, as you mentioned), successful multi-modal representations already exist (eg. VilBERT) and an assessment of their capability under different scenarios is a useful contribution. This is valid both in the case of a task dealing with multiple inputs, as in image captioning, or in the case of a “general” architecture handling a variety of different tasks separately (e.g., GPV https://arxiv.org/abs/2104.00743).
> However, we think that DNA and RNA are NOT different modalities: the human RNA “language” should be considered a subset of the human DNA language, in virtue of the fact that the mapping is 1:1. Tasks related to additional properties the RNA has compared to DNA (eg. folding of the single strand) have been excluded from our dataset, so it is safe to assume our tasks are working under a “unified” meta-language, the representations of which should belong to a single “modality”. If the reviewer deems this reasoning unsound or would like to point out any weaknesses, we are really interested in their opinion.
>
> We will improve our experimental section following the suggestions.

---

### Official Review · Reviewer_Bhcu · 2021-11-01

**Correctness:** 2
**Technical Novelty And Significance:** 1
**Empirical Novelty And Significance:** 3
**Recommendation:** 5
**Confidence:** 4

**Main Review:**

### Strengths

The authors do identify a gap in the literature and attempt to fill it with a set of complementary benchmarks. The datasets cover a variety of biological functions and organisms. The neural network baselines they train and evaluate seem to be generally done correctly. I appreciated the discussion on why some datasets were not included.

### Major weaknesses

The authors justify the need for deep learning benchmarks in the introduction by claiming that they can be faster and more robust than sequence-similarity or search-based methods. However, they do not run any of these methods as baselines on their benchmarks. For example, methods that build an alignment and then infer a tree should be very competitive on the hierarchical classification tasks. The authors mention that bioinformatics methods are often used as a gold standard, but do not mention what methods were used to label their datasets or compare their results and runtimes with neural network based methods.

At the end of the introduction, no justification is given as to why human DNA, bacterial 16S genes, and nematode ITS2 genes constitute different languages, or why in that case we can later expect pretraining on all of them simultaneously to be beneficial (and indeed, in most cases it’s not!). I recommend leaning less on the analogy to language, as there’s not a good 1-to-1 comparison between different languages and the genomes of different species, and the analogy confuses more than it helps. It's also not described how they are unified for combined pretraining: the authors should explicitly state this.

The biological background is often unclear and sometimes wrong. In the first paragraph of section 3, too strong a distinction is made between how DNA base pairs but how RNA merely "folds in different ways": RNA also base pairs, just not into a double helix. Furthermore, no mention is made of RNA base pairing to DNA. In the second paragraph of section 3, chromatin is only found in eukaryotic cells, not all cells as the text implies. The definitions for genomics, transcriptomics, and proteomics vs metagenomics, metatranscriptomics, and metaproteomics is also wrong. Genomics is simply the study of genes and genomes, whether limited to single organisms or not. By the definition given, deep mutational scanning would count as metagenomics, since the sequences come from multiple single-celled organisms! Finally, there is some debate about whether 16S and other single-gene studies count as metagenomics. Taken together, this is a major weakness for the paper. However, I think it should also be relatively easy to correct the terminology.

From the appendix, I think the authors use random or almost-random train/test splits for each task. Standard practice in biology is to avoid having highly similar sequences between train and test, as this can lead to over-estimating generalization performance. See, for example, these papers: https://www.nature.com/articles/s41580-019-0176-5, https://arxiv.org/abs/2006.16189.  In human genomics, standard practice is to split by chromosome: the authors claim that this would bias the results without providing any evidence or further justification. For hierarchical classification, this may not be as important; however, I suspect that simple sequence-similarity-based methods are both efficient and accurate for these. The paper would be improved by using biologically-justified splits, with explanations of why those splits were chosen.

In NLP (eg BERT) and in other work on pretraining for biological sequences (eg TAPE, ESM), the pretraining step is performed on many more sequences than are present in the final task. Is there any reason to believe that pretraining on the same sequences used as inputs in the final task should be generally helpful? It appears that they are, but the paper would be much stronger if we had an idea of why they are. Could it be that the LF-NO and BiLSTM-NO models in Table 1 have not converged?



### Minor weaknesses

Is it really necessary to split into kmers and then use byte-pair encoding? Why not just use the original 4-letter alphabet? To be clear, a sentence justifying the byte-pair encoding is sufficient to address this.

In the introduction, the authors should mention (and cite) recent work on using neural networks to understand glycans, in addition to DNA, RNA, and proteins. See, for example, this [paper](https://www.biorxiv.org/content/10.1101/2020.01.10.902114v1.abstract) and follow-ups.

I would have appreciated a figure describing the tasks.

Is BLAST really too slow for cancer diagnostics? Or did the author have some other search strategy in mind when they claim that database search is too slow in the intro?

Machine learning is not necessarily "computationally inexpensive," as the authors claim in the introduction.


**Summary Of The Paper:**

The authors identify a need for standard ML benchmarks in genomics in order to bridge the gap between the bioinformatics and deep learning communities and to make it easier for ML practitioners to contribute to genomics as they have in computer vision and natural language processing. The primary contributions of the paper are
- six curated, standardized supervised learning tasks involving sequences of nucleic acids
- baseline neural network models evaluated on those baselines


**Summary Of The Review:**

Although the paper proposes a useful resource for the genomics ML community, there are serious issues with its presentation of the biology, the data splits, and the baselines evaluated that prevent me from recommending acceptance as is. I do believe that these are addressable (although redoing splits and running baselines may be a tall order during the rebuttal period), and am open to revising my score.

---

> ### Author Response · Authors · 2021-11-19
> **.**
>
> We thank the reviewer for the helpful review. Since some reviewers asked us to run the experiments in a different setting, we are not able to submit a revised version of the manuscript in time. Nonetheless, to help us improve our work, we would appreciate it if the reviewer would take the time to read our rebuttal and comment on our answers.
>
> We will improve our work by adding more details regarding the techniques that have been used to compute the ground truth labels in the various tasks and shift the background motivations to more robust ones.
>
> Regarding the DNA/RNA to language analogy, we do believe it can provide additional “toolboxes” to handle problems that are defined on top of sequences and, hopefully, foster beneficial cooperation between NLP and Bioinformatics practitioners. It is important firstly to remember that NLP itself is not based directly on linguistics, but instead on statistical relations between entities, so “no” analogy will ever be 1-to-1, while still being useful to some degree. Secondly, we believe that “boundaries” between languages cannot be formally defined in this context, but an intuitive and pragmatic approximation can still be useful to guide the choice towards some tools instead of others. Phylogeny tasks are a good example for this intuition: if they are considered a single language one would be compelled to treat different samples as synonyms (and therefore choose features such as neighboring “words”), if however the tasks are considered under a multi-language framework, researchers would opt to go for (linguistic) phylogeny features (such as similarities in the stem of each word) and in fact this is the approach already used in (biological) phylogeny. Interestingly, in QIIME2 (a popular framework for 16S analysis), both intuitions are used on the same input features: the feature classifier is naive Bayes, where sequences are split into k-mers and the relevant feature is the co-occurrence of multiple k-mers, while the phylogenic metrics are computed on a SEPP (https://www.worldscientific.com/doi/pdf/10.1142/9789814366496_0024) tree.
>
>
> We will improve the work implementing your corrections on the background and terminology, moreover we will re-run our experiments using the suggested setting.
>
> We thank the reviewer for the references about tasks on glycans. It is an outstanding result, but, out of curiosity, would not graph neural networks be able to provide a stronger inductive bias, compared to NLP on sequences of “glycowords” given their branched nature?
>
> While we know practitioners who complained about BLAST speed in cancer diagnosis, we can not back this claim with proper scientific evidence, therefore we will remove the claim.

---

> > ### Comment · Reviewer_Bhcu · 2021-11-22
> > **Response to authors**
> >
> > I look forward to seeing the improved version of this paper with new experiments and clearer background and motivation!
> >
> >
> > >We thank the reviewer for the references about tasks on glycans. It is an outstanding result, but, out of curiosity, would not graph neural networks be able to provide a stronger inductive bias, compared to NLP on sequences of “glycowords” given their branched nature?
> >
> > Absolutely! The same researcher did that [too](https://www.sciencedirect.com/science/article/pii/S2211124721006161)!
> >
> > >Regarding the DNA/RNA to language analogy, we do believe it can provide additional “toolboxes” to handle problems that are defined on top of sequences and, hopefully, foster beneficial cooperation between NLP and Bioinformatics practitioners. It is important firstly to remember that NLP itself is not based directly on linguistics, but instead on statistical relations between entities, so “no” analogy will ever be 1-to-1, while still being useful to some degree.
> >
> > I agree that it's a good high-level analogy. We just need to be careful about taking it too far.
> >
> > > Secondly, we believe that “boundaries” between languages cannot be formally defined in this context, but an intuitive and pragmatic approximation can still be useful to guide the choice towards some tools instead of others. Phylogeny tasks are a good example for this intuition: if they are considered a single language one would be compelled to treat different samples as synonyms (and therefore choose features such as neighboring “words”), if however the tasks are considered under a multi-language framework, researchers would opt to go for (linguistic) phylogeny features (such as similarities in the stem of each word) and in fact this is the approach already used in (biological) phylogeny. Interestingly, in QIIME2 (a popular framework for 16S analysis), both intuitions are used on the same input features: the feature classifier is naive Bayes, where sequences are split into k-mers and the relevant feature is the co-occurrence of multiple k-mers, while the phylogenic metrics are computed on a SEPP (https://www.worldscientific.com/doi/pdf/10.1142/9789814366496_0024) tree.
> >
> > This is a much more nuanced than what was in the original paper. In general, I think the analogy to language is helpful for understanding the structure of the data, but this sort of nuance should be taken into account when deciding when and how to use the analogy.
> >
> > My concern about pretraining, however, is that only pretraining on a mix of sequences from the downstream tasks is probably insufficient. It would be analogous to only pretraining BERT on the SuperGLUE datasets. I recognize that pretraining a general DNA/RNA model on a large, diverse, subset of known genomic data is well outside the scope of this particular work. However, the paper would be clearer if it discussed this limitation.

---

### Official Review · Reviewer_QrMg · 2021-11-02

**Correctness:** 2
**Technical Novelty And Significance:** 2
**Empirical Novelty And Significance:** 2
**Recommendation:** 3
**Confidence:** 4

**Main Review:**

Pros:

- I applaud the authors for wanting to reach out to the broader ML community with problems in biology and bioinformatics.

- The authors did a good job of defining if a dataset was balanced or not.

Cons:

- Section 3 is a bit too long on it's own. I would have preferred if the biological topics were broken up into each of the respective subsections of 4.x; otherwise it is difficult to keep track of the biological relevance of each task.

- The selection of tasks could be further refined. Why not just focus on different chromatin annotation tasks, like 4.2? Why are there two taxonomy tasks? In particular, why was the its2 task chosen? (Also unclear what its2 is, and why it is an important task the ML community should focus on.)

- Ideally, this paper would present baselines from the original algorithms/packages cited for each of the respective works. e.g. for 4.2, how well does DeepSEA do?

- Dataset specific comments:

4.1 - The negative dataset doesn't seem to be GC content matched. This is typically controlled for in ML for DNA sequences. Moreover, promoters typically have many different transcription start sites. Which one are you picking? Using your negative selection strategy, your "negative" data most likely has a TSS in it too.

4.2 The "Objective" section is incorrect ("Determine all the possible ways a DNA strand can fold.") The annotations in the paper are DNAse1 hypersensitivity, TF binding, and histone methylation. While those are impacted by DNA shape, they are defined as 1D chromatin modifications. Descriptions of DNA folding should be reserved for HiC-like data. Moreover, since you recreated the train/test/validation splits, how are you controlling for homologous sequences? Perhaps a better split would be leaving out a chromosome?

4.6 Why not just use the GreenGenes algorithm directly? The accuracy of your algorithm is determined by the performance of GreenGenes chimera-calling. Does that have less of a greenhouse-emission footprint than a deep model? (As you mention in Section 7.)

Neutral:

Dataset 4.3 - Why was only mRNA chosen from all the different RNA subclasses? It seems like the database also contains non-messenger RNA data and annotations.



**Summary Of The Paper:**

The authors curate a set of machine learning tasks on non-protein biological sequences and provide baseline results using deep neural network language models.

**Summary Of The Review:**

While the authors have curated a number of non-protein biological datasets, I do not think the objectives of the tasks are refined enough, nor are the methods compelling enough, to justify publication.

---

> ### Author Response · Authors · 2021-11-19
> **.**
>
> We thank the reviewer for the helpful review. Since some reviewers asked us to run the experiments in a different setting, we are not able to submit a revised version of the manuscript in time. Nonetheless, to help us improve our work, we would appreciate it if the reviewer would take the time to read our rebuttal and comment on our answers.
>
> Since we designed this benchmark to evaluate general-purpose models, one of our priorities in choosing the tasks was to address many different settings in terms of the number of target classes and their distribution inside the data. We have favored this criterion rather than the actual impact of the task. Does the reviewer think this is a valuable motivation or that it would be better to have only high-impact tasks, even if they may be very similar in the setting?
> The ITS2 task differs from 16S in terms of the number of levels in the hierarchy and the number of samples for each class. In particular, it is a setting where few-shot learning approaches may be beneficial. Moreover, it is one of the few datasets with manually curated labels.
>
> For 4.1 we used EPDNew, so the positive samples contain the primary TSS, we will specify that negative samples may contain secondary TSS. Thank you for the suggestion on GC content checking.
>
> Regarding 4.2, we oversimplified the objective to the point it became incorrect, thank you for the corrections. We agree that held-out chromosomes are a more correct approach and we will re-do the experiments.
>
> Our focus is to provide a benchmark for the development of new general models, rather than provide a new state-of-the-art approach. Moreover, a task-specific baseline is already available only for 4.2 only: 4.1 and 4.3 are differently formulated compared to other works, 4.4 dataset was subject to label cleaning, 4.5 was not solved by any ML model we are aware of, and on 4.6 Bellerophon (a chimera checker) is already used to provide the ground truth by the curators of the GreenGenes database. Since we are interested in promoting general-purpose models and all these approaches are task-specific, we have decided to invest our computational resources in training new neural models that may be considered baselines for future works, rather than re-training existing baselines on our dataset. Does the reviewer think that, instead, we should focus on re-training previous baselines?
>
> Regarding 4.3, we decided to discard the miRNAs because they are significantly shorter than mRNA and the other ncRNA as they were too few in the RNALocate database to justify the inclusion, and we wanted our tasks to not be too different in terms of input sequences.

---

### Official Review · Reviewer_dxXz · 2021-11-02

**Correctness:** 2
**Technical Novelty And Significance:** 1
**Empirical Novelty And Significance:** 1
**Recommendation:** 3
**Confidence:** 4

**Main Review:**

Strengths:
- The tasks are fairly broad, ranging from regulatory genomics, to phylogeny, to technical biases.
- I appreciate how the authors try to make a connection with the ML tasks in biology to NLP.

Limitations:
- Motivation isn't clear: why is it desirable to have a single model that has mediocre predictive performance on all tasks, rather than a specialized model SOTA on each task. Tasks in genomics can be so diverse, why should we expect a single, universal nucleic acid representation that is good for all tasks? For instance, multi-cellular organisms carry the same DNA in each cell, but there is a different gene expression/regulation across different cell types. Moreover, there is a different, timed cellular processes, such as development, within each cell type. This spatial and temporal complexity makes me wonder whether it is sensible to expect that a single representation of nucleic acids would be suitable for a diverse set of prediction tasks in biology. Perhaps the authors can add a rationale for this given the complexity of biological systems.
- The tasks/datasets do not have enough information and also have inaccuracies.
	- For promoter detection, what are the positive label data and what are the negative label data? Which cell types do they come from?
	- Random splits are one strategy for generating a test set. Another, better option that is practiced in the field is held out chromosomes.
	- It appears that each dataset is preprocessed data from papers, but no clear citations are shown.
	- Chromatin classification task, is this the deepsea dataset? If so, the objective is wrong -- it is not about determining all the possible ways a DNA strand can fold! Moreover, the main metric is wrong -- AUC is not sensitive to class imbalance (which this dataset suffers from, assuming it is DeepSea). A better metric is AUPR.
- There is no intuition for the kinds of features that are important for each task. Regulatory genomic features are very different from phylogeny problems. A better understanding of the features helps design models with better inductive biases. For example, the bi-LSTM model that is used has practically no inductive bias for regulatory genomics tasks, which is why the AUC is 0.51! If the authors would describe the kinds of features that are important for each task, it would be clear that a convolutional network is more suitable for regulatory genomics, or at least applying a convolutional layer prior to the bi-LSTM can significantly boost performance -- this is a common architectural choice first proposed with DANQ.
- There were no comparisons to supervised models that are SOTA for each task. This would help give context to the performance of these pretraind language models.
- For pretrained models, the authors should benchmark against bigbird and DNABERT.  These models were pretrained across the human genome, not just for the task(s) at hand. It would at least be sensible to pretrain on the human genome, then tune the pretraining on the task at hand.
- Comments on text:
	- The second paragraph is an inaccurate portrayal of the field, with too much emphasis on homology search, which is not one of the benchmark tasks in this paper.
	- The longformer is not state-of-the-art -- it has yet to be established.
- It is not clear biology community would care if performance were better for these tasks. There should be a discussion of how labels are experimentally measured and are thus noisy. They are not ground truth. It's not clear what better performance gives after a certain point, because models may have all captured biology, but better performance may reflect a better model of technical biases or measurement noise. Thus other downstream evaluation metrics are needed, such as model interpretability (i.e. motif discovery and localization) or variant effect prediction, both of which are downstream use cases of NNs in regulatory genomics.
- IMO, a better sales pitch for this work is to emphasize representation learning versus the more traditional one-hot representation and supervised representation learning versus pre-training with self-supervised masked language modeling. Importantly, if the performance of the language models are not comparable to supervised models, then there will be no engagement between the NLP and bio community.

**Summary Of The Paper:**

This paper proposes a set of 6 benchmark tasks to assess the prediction performance of different models in genomics. The motivation is that there lacks standardized datasets and evaluation methods to seamlessly compare different models across different datasets due to the heterogeneity in the databases and the pre-processing required to get it in a form that is suitable for ML analysis. Therefore the goal is to introduce a set of diverse tasks for DNA/RNA to benchmark pre-trained models, similar to what TAPE has done for proteins. There are 3 tasks for regulatory genomics, 2 tasks based on phylogeny, and 1 task based on technical biases. The authors explore a few baseline models, including a Longformer and bi-directional LSTM, either trained directly on the task, pretrained on the task and then fine tuned on the same task, or pretrained on sequences across all tasks and fine tuned on each task individually. The prediction performance was compared using a metric that the authors thought was appropriate. This paper attempts to bridge the NLP community and the comp bio community by framing numerous comp bio tasks with analogies to NLP.

**Summary Of The Review:**

On surface, the presentation and motivation seems great (to a researcher outside of biology). But once you realize data/model performance are limited compared to existing supervised models, it may mislead the NLP community to focus on ill-formed problems and only provide marginal "relative" gains with no impact and no engagement with the bio community. There is a lack of clarity of the datasets, i.e. where they come from and basic intuition for the important features. The baseline models are poor choices for (regulatory) genomics, they are not common. Moreover, prediction performance is only one aspect of evaluation. In biology, evaluation needs to incorporate other aspects, such as model interpretability and variant effect predictions (of GWAS or eQTLs). While bridging both communities (NLP and comp bio) is important, this work seems too premature. I encourage the authors to take this feedback to guide them to generate a much needed set of benchmark tasks, baseline models, and better evaluations that would also be of interest to the comp bio community.

---

> ### Author Response · Authors · 2021-11-19
> **.**
>
> We thank the reviewer for the helpful comments. Since some reviewers asked us to run the experiments in a different setting, we are not able to submit a revised version of the manuscript in time. Nonetheless, to help us improve our work, we would appreciate it if the reviewer would take the time to read our rebuttal and comment on our answers.
>
> Biological systems are diverse, but also NLP tasks can be diverse: think for example at question answering, abusive language detection, image captioning, and argument mining. However, BERT and other language models can positively affect a wide range of tasks. Generalized benchmarks such as GLUE/SuperGLUE have been fundamental for such innovations. Similarly, we believe that similar general models may be beneficial in bioinformatics tasks, and therefore we propose a benchmark to facilitate the research of new state-of-the-art models. It would be helpful for us to know whether the reviewer considers this motivation convincing or if they think there are aspects that we may have neglected.
>
>
> Additional information on our data, their source, and how we have processed them are given in Appendix B. For the specific case of PromD, we use the entries of EPDNew, therefore input sequences are taken from the GRCHg38 reference genome. As a result, there is no single cell line from which they were taken, and the TSS coordinates used to annotate our dataset correspond to the main TSS site.
>
> Thank you for the corrections about DeepSea. We had chosen AUC following the authors of BigBird, but we will re-do the experiments considering AUPR and biologically-motivated splits.
>
> One of our main goals is to pave the way to new general approaches, therefore we proposed only simple baselines and did not focus on exploring which could be the best set of features nor the best model. Regarding downstream interpretability, we consider them out of scope, since they are task-specific and they greatly increase the domain knowledge required to solve a task (one of the problems we are trying to mitigate). If the reviewer does not agree with this claim, we would like to discuss this point further.

---

### Official Review · Reviewer_wY3Y · 2021-11-02

**Correctness:** 3
**Technical Novelty And Significance:** 1
**Empirical Novelty And Significance:** 3
**Recommendation:** 5
**Confidence:** 4

**Main Review:**

The manuscript is generally clear and well written. Building a
standardized benchmark for comparing methods on prediction tasks from
DNA or RNA is a useful contribution.

I am not sure however that the proposed contribution is significant
enough for a publication at ICLR. First, I am not immediately
convinced by the importance of comparing models on such a large
variety of tasks. The problems listed in the proposed benchmark are
indeed very different, and may call for radically different
approaches. In other words, it is not clear to me that one should aim
for a method that predicts well on all these tasks. To me a more
relevant benchmark could focus on a more narrow problem (eg promoter
or chromatin feature prediction). I may be wrong but I think it would
be useful to motivate this point a bit further.

Of note, the kipoi repository (https://kipoi.org/) already partially
addresses this task, and should be included in the discussion. More
generally, the description of the field could be made more thorough or
detailed at some places. For example:

- I disagree with the claim that "Umarov & Solovyev (2017) were the
  first to train a neural network on promoter detection" when DeepBind
  (Alipanahi et al., 2015) tackled the same problem.

- It could be useful to say a bit more about the longformer model that
  is used as a baseline, and about contextual embeddings, and why
  either are expected to help on the tasks chosen in the benchmark. On
  a related point, I was surprised by the large sequence size used for
  the promD task (2048), while this specific task is often done on
  shorter (a few hundreds of bp) sequences, where long range models
  are typically less useful.

- ChromC seems to rely on the task tackled in the DeepSea (Zhou and
  Troyanskaya, 2015) paper, where the goal is to predict chromatin
  features such as transcription factor binding, histone or DNAse
  features, not DNA folding.

- I would need more explanations on the claim that "Unlike natural
languages, DNA and RNA sequences are not “naturally” split into words,
therefore it is necessary to define synthetic tokens.". A common
approach (taken by both deepbind and deepsea) is to directly start
from a one-hot encoded sequence. Working at the kmer level can improve
the performance but is not a necessary step.


**Summary Of The Paper:**

This contribution introduces a benchmark for the evaluation of neural
networks for supervised classification of DNA and RNA molecules. The
benchmark consists of six datasets covering DNA and RNA as well as
different learning tasks (binary classification, multiclass,
hierarchical), sequence lengths and sample sizes. The authors propose
baselines on each of these tasks, one based on an LSTM, the other on a
recent transformer architecture.


**Summary Of The Review:**

While a benchmark for prediction from DNA/RNA is a useful contribution, I find the following main limitations:
1 - Limited contribution (given the scope of ICLR).
2 - Not sufficiently justified use of a common benchmark for very diverse tasks.
3 - Sometimes not detailed/justified enough elements given the field.

---

> ### Author Response · Authors · 2021-11-19
> **.**
>
> We thank the reviewer for the helpful review. Since some reviewers asked us to run the experiments in a different setting, we are not able to submit a revised version of the manuscript in time. Nonetheless, to help us improve our work, we would appreciate it if the reviewer would take the time to read our rebuttal and comment on our answers.
>
> Although radically different tasks indeed require different approaches (and in general it should be true that a task-specific solution performs better than a general one), one of our main objectives is to promote the exploration of novel, hopefully general, approaches, akin to how GLUE/SuperGLUE promoted the use of unified architectures for a broad class of natural language tasks. We believe that these benchmarks greatly contributed to the transformer’s popularity in NLP and that a field such as bioinformatics could benefit similarly from a multi-task assessment. It would be helpful for us to know whether the reviewer considers this motivation convincing or if they think there are aspects that we may have neglected.
>
> We were not aware of Kipoi and we will mention it in future versions of this work. It is interesting, although the main goals seem to differ from ours: they offer a ready-to-use set of pre-trained models and a unified interface to access/retrain them, to promote quick deployment and lower the effort required for Bioinformatics practitioners in entering the ML field (in contrast to our goal of promoting the interaction between NLP/ML practitioners and Bioinformatics ones, and of evaluating new approaches in a unified way). Furthermore, as they mention, they facilitate the comparison between different models, but how these models are compared is left to the researchers. We believe both Kipoi and a BANANA/TAPE-like benchmark are useful tools but at different stages of the development of a Bioinformatics model. The former aims to evaluate how existing models and techniques perform on a single, well-defined, and specific task and then choose the best. The latter aim to assess their ability to generalize and to be applied and re-used in different contexts and tasks.
>
> The benchmark we proposed forces the use of neither contextual embeddings nor character-wise approaches, but it is instead compatible with both. That said, the application of NLP methods to bioinformatics is not new, think for example the use of CNN on characters. Since in the last decade all the focus in NLP language models has been on contextual embeddings, we believe that the present benchmark should be designed primarily to test state-of-the-art, and in particular contextual embeddings-based NLP, architectures.
>
> Although less useful in practice, a lengthy input for PromD should address and help investigate whether the lack of “utility” for long-range models is a characteristic of the data (eg. the presence of minor TSS sites, as hinted by reviewer dxXz, could be a problem in longer sequences), or a limitation of the models (eg. an LSTM may struggle to keep the relevant information in memory for too long, while a “traditional” transformer, such as BERT, always has access to the full sequence). As a side note, appendix F of BigBird paper (accessible in the pre-print version) mentions promoters of length 8000bp for the same task. Similar figures in the published version hint at the fact that they did not repeat the experiments on shorter lengths for their final manuscript.
>
> ChromC is exactly the task addressed in the DeepSea paper, except for the input being “reverted” from one-hot to sequences and splits randomly reshuffled to uniform splits (again with the same design principle of mirroring its dual, mRNAC).
> Thank you for your corrections, in our effort to make the paper accessible to a mixed audience of NLP and Bioinformatics practitioners we have oversimplified some concepts introducing errors.
>
> The statement about the “necessity of synthetic tokens” refers to the specific architectures we tested, it was not meant to be a general statement. It is common practice to use transformers with subword tokens, such as those produced by byte-pair encoding, because this solves the problem of out of vocabulary words, “automatically” encodes features related to token frequency in the corpus, and reduces the input dimensionality. This is important for memory-hungry architectures such as transformers, but clearly, a CNN would not need such dimensionality reduction.
> However, the BPE algorithm works better on a sequence of multiple smaller words, compared to a single (possibly prohibitively) long “word”, so a k-mer split is a way to solve this problem. BigBird (Zaheer et al., 2020, preprint version) proposed a different word-splitting procedure through random length splits, in a preliminary evaluation we realized performance improved a bit with their method, but the increased computational overhead required did not justify this approach for our tests (also considering our limited hardware setup).

---

> > ### Comment · Reviewer_wY3Y · 2021-11-29
> > **Answer to the rebuttal**
> >
> > Dear authors,
> >
> > Thank you for for taking the time to provide such a detailed answer.
> >
> > I agree that multi-task assessment can help recognize Transformer-like disruptive approaches that yield large improvements across many related problems. I just think that this will be a small minority of cases. That being said, it doesn't hurt to offer benchmark for all important problems.
> >
> > In the view of your answers I will raise my score. However I feel like too much work would be required for this article to be published in its current form, both on its formulation and the content of the benchmark. I hope the comments that were made by all reviewers will help improve this work, as such a benchmark would be a very useful contribution.

---

### Decision · Program_Chairs · 2022-01-20

**Decision:**

Reject

**Comment:**

While all reviewers applaud the motivation to bridge the gap between machine learning and bioinformatics communities, they also raise a number of concerns regarding the choice of tasks and of baselines, and about the accuracy in their description. They feel the paper is not ready to be published in its current form, and we hope that their comments will help the authors prepare a revised version for the future.